# SARS-CoV-2 Infection in Captive Hippos (*Hippopotamus amphibius*), Belgium

**DOI:** 10.3390/ani13020316

**Published:** 2023-01-16

**Authors:** Francis Vercammen, Brigitte Cay, Sophie Gryseels, Nadège Balmelle, Léa Joffrin, Koenraad Van Hoorde, Bavo Verhaegen, Elisabeth Mathijs, Rianne Van Vredendaal, Tanmay Dharmadhikari, Koen Chiers, Tim J. S. Van Olmen, Gianfilippo Agliani, Judith M. A. Van den Brand, Herwig Leirs

**Affiliations:** 1Centre for Research and Conservation, Antwerp Zoo Society, 2018 Antwerp, Belgium; 2Unit Enzootic, Vector-Borne and Bee Diseases, Sciensano, 1180 Brussels, Belgium; 3Department of Biology, Antwerp University, 2610 Antwerp, Belgium; 4Directorate Taxonomy and Phylogeny, Royal Belgian Institute of Natural Sciences, 1000 Brussels, Belgium; 5Unit Foodborne Pathogens, Sciensano, 1050 Brussels, Belgium; 6Veterinary Pathology, Department of Pathobiology, Pharmacology and Zoological Medicine, Ghent University, 9820 Merelbeke, Belgium; 7Division of Pathology, Faculty of Veterinary Medicine, Utrecht University, 3584 CL Utrecht, The Netherlands

**Keywords:** captive, faeces, hippo, nasal, pool, SARS-CoV-2

## Abstract

**Simple Summary:**

We report the first world case of SARS-CoV-2 in captive hippos. Two adult animals were naturally infected in November 2021. Both had snot for a few days but did not need treatment. The virus was found in their noses, faeces and pool water. Antibodies were also found in the bloodstream.

**Abstract:**

Two adult female hippos in Zoo Antwerp who were naturally infected with SARS-CoV-2 showed nasal discharge for a few days. Virus was detected by immunocytochemistry and PCR in nasal swab samples and by PCR in faeces and pool water. Serology was also positive. No treatment was necessary.

## 1. Introduction

SARS-CoV-2 is the cause of the human pandemic disease COVID-19, and this virus can infect several domestic and wild animal species [1,2,3]. Domestic cats, dogs and ferrets do not fall severely ill, but farmed mink can die [4]. Wild free-living white-tailed deer in the United States appear to be a reservoir for this virus after several introductions from humans, but whether or not they develop signs and disease remains unknown so far [5,6]. Infected captive wild animals are mainly felines without serious signs, including lions, tigers, pumas, fishing cats, Canada lynxes and snow leopards, although a few Asiatic lions and snow leopards have died. In addition to captive wild felines, SARS-CoV-2 can also infect captive gorillas, hyenas, binturongs, Asian small-clawed otters and coatis [1,2,3]. We report the first SARS-CoV-2 case in captive hippos in Zoo Antwerp in November 2021.

## 2. Materials and Methods

### 2.1. Animals

The common hippo (*Hippopotamus amphibius*) is listed as vulnerable by the International Union for Conservation of Nature with a population of approximately 115,000–130,000 animals in sub-Saharan Africa. The primary threats to common hippos are habitat loss or degradation and illegal and unregulated hunting for meat and ivory, which is found in the canine teeth [7]. Zoo Antwerp participates in the European Association of Zoos and Aquaria ex situ population management programme (EEP) and is home to 2 adult females: hippo A and B are 41 and 14.5 years old and weigh about 1600 and 1300 kg, respectively. Both hippos developed bilateral mucopurulent nasal discharge on 23 November 2021 for a few days. At that time, they were already housed indoors for the cold winter. There were no other symptoms, but as we had not seen any such discharge in any of our hippos before, samples were sent for bacteriological culture. The laboratory result was negative for pathogenic bacteria and then PCR was performed to rule out the possibility of SARS-CoV-2. No treatment was necessary.

### 2.2. Samples and Methods

Overall, 20 and 15 nasal samples were obtained from hippo A and B, respectively, between 23 November and 31 December 2021, and smears were immediately stained with Diff-Quik for cytology. The same nasal swab smears were also stained with antibody against the nucleocapsid protein [8], and positive mink lung tissue was used as control. Briefly, smears were fixed in acetone, endogenous peroxidase and non-specific antibody binding were blocked, primary antibody (human SARS coronavirus nucleoprotein antibody 40143-T62, Bio-Connect, Huissen, The Netherlands) was incubated and secondary antibody (BrightVision Poly-HRP-Anti Mouse/Rabbit IgG, VWR, Leuven, Belgium) was incubated; finally, 3-Amino-9-Ethylcarbazole, which produces a red end-product and, subsequently, haematoxylin counterstain were applied, and a coverlid was mounted with an aqueous mounting medium (Aquatex, VWR, Leuven, Belgium).

Then, 25 and 18 other nasal samples were obtained from hippo A and B, respectively, between 23 November 2021 and 4 January 2022 for PCR, and 5 of these were sequenced. Total RNAs were extracted using Indimag Pathogen kit and Indimag 48S robot for automated magnetic bead-based extraction of viral RNA. A positive extraction control of human positive samples was always extracted simultaneously. For the extraction control, the viral gene N and the human RNASE P endogenous gene were detected. A reverse transcription quantitative PCR (RT-qPCR) assay was adapted from the 2019 nCoV CDC Kit targeting N1 [9]. As internal qPCR control, the hippopotamus beta-actin gene was amplified using the ß-actin Control kit Yakima Yellow-Eclipse Dark Quencher (Eurogentech, Cat. RT-CKYD-ACTB, Seraing, Belgium). As amplification control, a diluted plasmid carrying SARS-CoV-2 N gene was used as template for gene N detection. Reactions were incubated at 50 °C for 10 min and 95 °C for 3 min in order to conduct reverse transcription of viral RNA, samples denaturation and enzyme activation. These steps were followed by PCR amplification including 45 cycles at 95 °C for 15 s and 58 °C for 30 s and a final cooling step at 40 °C for 10 min. Results were considered valid only when the cycle threshold (C_t_) value of the beta-actin reference gene was ≤37. C_t_ values ≤ 40 of the nasal samples were considered positive. The sequencing of the SARS-CoV-2 genome was performed according to the Midnight PCR tiling protocol developed by the ARTIC network and collaborators with minor modifications [10]. Briefly, purified RNA was reverse transcribed using Lunascript RT Supermix Kit (New England Biolabs, Leiden, The Netherlands) according to the instructions. The non-overlapping ‘midnight’ 1200 bp PCR reactions were performed in 2 separate reaction tubes. Cycling conditions were as follows: 98 °C for 30 s, 25–35 cycles of 15 s at 98 °C, 2.5 min at 65 °C and, finally, 2 min at 72 °C. Amplicons were purified using the Agencourt AMPure XP system (Beckman Coulter, Suarléé, Belgium) and quantified on a Qubit 4 fluorometer (ThermoFisher Scientific, Erembodegem-Aalst, Belgium). Amplicons were pooled in an equimolar manner prior library preparation using the Ligation Sequencing Kit and the Native Barcoding Expansion 1–12 Pack (SQK-LSK109–EXP-NBD104; Oxford Nanopore Technologies, Oxford, UK) according to the manufacturer’s recommendations. Sequencing was performed on a Mk1B MinION (Oxford Nanopore Technologies) using MinKNOW software v21.06.0 with ‘basecalling’ and ‘demultiplexing’ options enabled (guppy v5.0.11) for 24 h on a Flongle Flow Cell (FLO-FLG001; Oxford Nanopore Technologies). Consensus sequences were generated from the barcode-sorted, quality-filtered FAST5 and FASTQ files output files, using the ARTIC bioinformatics pipeline [11]. The SARS-CoV-2 lineage was determined according to the Pango nomenclature using the pangolin tool v3.1.17 [12]. The 29,700-nucleotide-long sequence hCoV/hippo/Belgium/U21158510003/2021 was submitted to the public database GISAID (accession ID: EPI_ISL_7328236).

When the animals defecated on land, 23 individual faecal samples were obtained from 15 December 2021 until 11 January 2022 for PCR. Faeces was rinsed or mixed with phosphate-buffered saline and added to AVL for RNA extraction with the QIAmp Viral RNA extraction kit (Qiagen, Hilden, Germany). The CDC N1/N2 real-time RT-PCR diagnostic primer-probe panel [13] and the Luna Universal Probe One-Step RT-qPCR Kit (NEB, Ipswich, MA, USA) were used for RNA testing. The RT-PCR was run on a StepOne™ Real-Time PCR System (Applied Biosystems, Thermo Fisher Scientific, Brussels, Belgium) using the following programme: 52 °C for 10 min, 95 °C for 2 min and 45 cycles of 95 °C for 10 s and 55 °C for 30 s. C_t_ values ≤ 40 were considered positive.

Seventeen pool water samples were analysed with PCR from 15 December 2021 until 1 February 2022. In total, 50 mL of pool water was centrifugated, and the supernatant was transferred to a centrifugal filter Centricon Plus-70 100 KDa (Merck Life Science, Overijse, Belgium) and concentrated by additional centrifugation. RNA was extracted by using the Maxwell RSC PureFood GMO and Authentication Kit (Promega, Leiden, The Netherlands), according to the manufacturer’s instructions and the Maxwell Rapid Sample Concentrator (RSC) instrument. RT-qPCR detected N1, N2 and E [13,14]. C_t_ values ≤ 40 were considered positive. Standard curves of the 5-fold serial dilutions of SARS-CoV-2 RNA (NIBSC code 19/304) were utilised. For each sample, the SARS-CoV-2 RNA concentration was calculated from the standard curve and the different concentrate and eluate volumes. The amplification was performed under the following conditions: one cycle at 50 °C for 5 min (RT) and one cycle at 95 °C for 20 s (Taq polymerase activation); 45 cycles at 95 °C for 5 s (denaturation) and at 60 °C for 30 s (annealing/elongation); RT-qPCR assays for N1 and N2 were multiplex, while RT-qPCR assay for E was singleplex.

A total of 28 and 23 saliva swab samples were obtained from hippo A and B, respectively, over a period of 2 months. Since there was no need to sedate or anaesthetise the animals for medical treatment, an intravenous blood draw was impossible. However, hippo A was a docile animal and after cleaning its mouth with running water and drying the gingiva at the base of a tusk with sterile gauze, a small surgical scalpel incision was made without any response from the animal. Blood was drawn into a needleless syringe and transferred into a serum tube on 1 February 2022. The bleeding stopped without intervention within minutes. Archived sera from 2 other hippos in 2012 and 2016, long before the pandemic, were used as negative controls. Two different commercial ELISA tests were used in serology according to the manufacturer’s instructions: ELISA Wantai based on the receptor-binding domain [15], and ID Screen ELISA multi-species based on the nucleocapsid. To confirm the specificity, the same samples were analysed with the Bovine Coronavirus Monoscreen Ab ELISA. The ELISA Wantai was used for detecting salivary antibodies.

## 3. Results

### 3.1. Interpretation of the Cytology and the Immunocytochemistry

Both hippos showed similar cytology. Only during the first days of the nasal discharge was much mucus observed, which then decreased to a normal small mucus amount. Large polygonal cells with a large amount of cytoplasm and a central nucleus (squamous epithelial cells) were always present. Smaller oval to round cells with little cytoplasm and a large nucleus (mononuclear cells) were intermittently present.

Virus was detected in both hippos in the first samples. Positive immunolabelling showed red staining in the cytoplasm of mononuclear cells (Figure 1A). The squamous epithelial cells were negative (Figure 1B).

### 3.2. Analysis of Nasal Swabs with PCR and Sequencing

Hippo A and B were positive in 12 of 25 (C_t_ values 15.63–40.0) and in 7 of 18 (C_t_ values 26.17–38.2) samples, respectively (Figure 2).

This virus belongs to lineage AY.125 or B.1.617.2.215 (a European lineage of the Delta variant). Blast analysis on 9 December 2021 on GISAID showed that the virus shared 99.98% nucleotide identity with a Delta variant detected in Belgium in October 2021 (EPI_ISL_5507546). Three single nucleotide polymorphisms and two single nucleotide deletions distinguish both viruses.

### 3.3. Analysis of Faeces and Pool Water with PCR

Hippo A and B had 3 and 2 positive samples out of 23 faecal samples, respectively (Figure 2, Table 1). Nine (53%) pool water samples were positive, of which seven, six and three samples were positive for N1, N2 and E, respectively (Table 1). Only two pool water samples showed a positive result for all three target genes, while three samples were positive for two target genes and four samples for one target gene.

### 3.4. ELISA Analysis of Saliva and Serum Samples

Antibodies were detected in one single saliva sample of hippo A, but with much lower reactivity than in serum. All other saliva samples were negative. The serum of hippo A was positive in Wantai (16.79; cut-off 1.00) and in ID Screen (60.33%; cut-off 60.00%). All three serum samples were negative in the Bovine Coronavirus Monoscreen Ab ELISA.

## 4. Discussion

The properties of 25 important amino acids in the angiotensin-converting enzyme 2 (ACE2) receptor protein for binding the spike protein classified hippos with a medium binding score (20/25 amino acids identical to human binding residues) [16]. Combining species traits with three-dimensional modelling of host–virus proteins using machine learning predicted a low 0.44 susceptibility score, whereas the crab-eating macaque had the highest score of 0.99, and the Cape golden mole had the lowest of 0.02 [17]. In addition to viral entry, other factors play a role in determining an infection or disease, such as population density and airflow and ventilation [18,19]. However, our two hippos lived alone in the building, which has an air volume of about 2150 m^3^ with mechanical ventilation and their pool contained city tap water. It was unlikely that SARS-CoV-2 would be found in this species. The shortest distance between visitor and hippo is 2.5 m, but zookeepers get closer for training and inspection, although none of them showed any sign of infection before or after the first hippo’s nasal discharge. Zookeepers were vaccinated, but could have been asymptomatic carriers, similar to what has happened in other zoos [20,21,22,23,24]. Although the animal feed may have been contaminated by the zookeepers, indirectly transmitting the infection, the feed was not examined, but the risk of contracting COVID-19 in this way is considered quite low even in humans [21]. The national incidence of 1705 per 100,000 inhabitants with a 99.6% Delta variant presence at that time and the small genetic difference strengthens the supposition that the virus was most likely transmitted from humans to animals.

Hippos are semi-aquatic mammals and spend much time in water with a constant exchange of nasal fluid. Fifty-three percent of water samples were PCR-positive contrary to only 27% of the combined nasal and faecal samples. Monitoring pool water, like wastewater, appears useful for detection of viral dynamics even if it is most likely not infectious virus [25,26]. The risk of transmission from the hippos is not clear, because viral isolation was not performed. Captive lions, tigers and gorillas that did not show severe disease while shedding the virus probably transmitted the virus to their conspecifics, but this could not be proven [20,21,22]. Wild free-living white-tailed deer had substantial deer-to-deer transmission [5]. The risk of infected wild zoo animals shedding enough virus to infect zookeepers or visitors is considered even lower compared to pet animals by several national health agencies, FAO and OIE [3].

Detection of virus in nasal cells and antibodies in serum suggests that a real infection has taken place. However, there are some limitations to this study. Other useful techniques for detecting inflammation, such as rhinoscopy, radiography or ultrasonography, were not performed due to the physical proportions of these large heavyweights and the need for anaesthesia. Nor were nasal washes performed to obtain infected cells for virus isolation by culturing and monitoring the cytopathic effect [21,22,23,27].

Why only nasal discharge was present without other signs of respiratory infection could be due to the difference in the receptor ACE2 expression across tissues, as in lion, cheetah and lynx [28]. While recognition of the receptor ACE2 is a key determining factor for cross-species infection of SARS-CoV-2, the transmembrane serine protease 2 (TMPRSS2) is also important to cleave the spike protein and facilitate viral–host membrane fusion. Both ACE2 and TMPRSS2 are necessary for infection of the cells [29,30]. Future research should also focus on studying mechanisms underlying differences between species in disease presentation and severity.

## 5. Conclusions

This is the first world case of SARS-CoV-2 in hippos showing only nasal discharge. The virus was found in nasal swabs, faeces and pool water. The Delta variant was at that time present with a very high national incidence. The slight genetic difference between the human and the hippo viruses suggests that humans infected the hippos.

## Figures and Tables

**Figure 1 animals-13-00316-f001:**
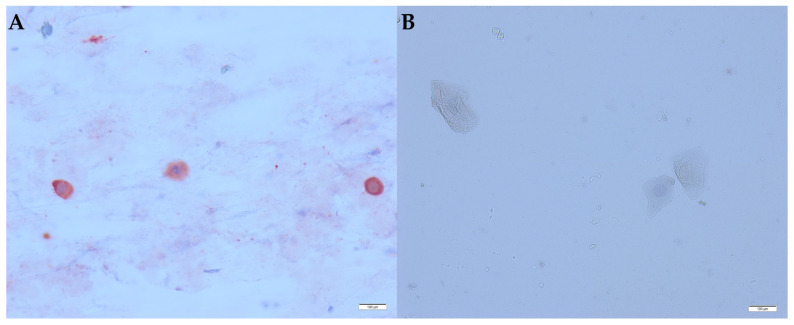
Immunocytochemistry of a nasal swab smear of a hippo. Magnification 40×. (**A**) Hippo with SARS-CoV-2 antigen in mononuclear cells; (**B**) hippo without SARS-CoV-2 antigen in squamous epithelial cells, bar = 100 µm.

**Figure 2 animals-13-00316-f002:**
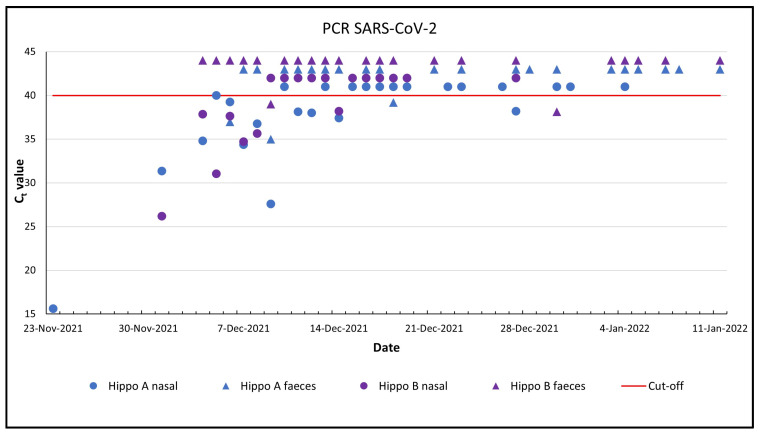
PCR detection of SARS-CoV-2 in nasal swabs and faeces of two hippos. For visual clarity, the negative samples (C_t_ > 40) were given a fictitious number: 41, 42, 43 and 44.

**Table 1 animals-13-00316-t001:** PCR C_t_ results of 23 faecal and 17 pool water samples.

Source	Faeces		Pool Water		
	Number of Positives	C_t_ values	Gene	Number of Positives	C_t_ values
Hippo A Hippo B Pool Water Pool Water Pool Water Pool Water	3 2	35.0–39.2 38.1–39.0	N1 N2 E Total	7 6 3 9	33.56–35.78 34.78–38.09 36.71–37.54

C_t_: cycle threshold.

## Data Availability

The data supporting the findings of this study are included within the main document and are available upon reasonable request.

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
