# Peer review of "SARS-CoV-2 Infection in Captive Hippos (Hippopotamus amphibius), Belgium"

_animals, 2023, doi:10.3390/ani13020316_

Round 1
Reviewer 1 Report
Comments on the reviewed manuscript “SARS-CoV-2 Infection in Captive Hippos (Hippopotamus amphibius), Belgium” submitted to the Animals
General comments
I appreciate the opportunity to review this impressive manuscript, which is a case report on SARS-CoV-2 infection in captive Hippos.
The manuscript is well written, explaining in detail the entire course of the disease, diagnostic methods, and outcome of the case.
The topic is original and important for understanding still obscure aspects of the complex epidemiological chain, etiology, and pathogenesis of SARS-CoV-2 infection in animals.
I consider the methodology well written and detailed, although it is in an unusual form of topics and short sub-topics, marked with dots.
The conclusion is coherent with the results and the discussion.
References are within the scope of the article.
The two figures are aptly illustrating the manuscript.
Congratulations to the authors.
Reviewer 2 Report
The Authors report in a short format MS the infection with SARS-CoV-2 of two captive hippos in Belgium. The data are relevant but I was a bit disappointed with the style/quality of manuscript writing. For instance, the introduction is very short and does not provide a sufficient backgrund.
Also, at point 2.2, 2.3, 2.4 , 3.1 to 3.7 appear as one-shot sentences.
Please note that point 3.2 heading is not formatted as a subheading whislt point 3.4 is duplicated.
Thea headings/subheadings should be a bit more articolated. For isntance "PCR pool water" should read "Analysis of pool water" or similar.
Overall, my impression is that this was a quick transcription of records from a book/registry rather then a digested manuscript.
Reviewer 3 Report
Dear authors,
thank you so much for the short and very comprehensive case presentation of captive Hippos showing cells positive for SARS CoV-2 as well as viral RNA in the faeces, water and serum positivity. The manuscript is a great contribution to the knowledge of wide spectrum of species showing positive results lacking severe clinical signs.
Overall, the study is great and well written.
Can I kindly ask the authors to address the following aspects please?
1.) line 152 Detection of virus in nasal cells and antibodies in serum, demonstrates that a real infection has taken place: this can be true, however, there are some aspects which also argue against a strong sentence like this and please add some aspects in the discussion, which allow the reader to get insights into a critical discussion. I envision a mini paragraph with few sentences addressing "Limitations" of the study.
2.) please include some information on the potential risk of transmission of the virus of species like these hippos and also other info from the literature on other species, which lack severe clinical presenation and are shedding virus positive cells.
3.) please fuse the information of lines 89 and 90
4.) is there a chance to amend the text of samples and methods to avoid bullet points and write a short text instead please?
5.) lines 117 to 134: would it help to summarise the information given in these lines in a table for a better overview for the reader?
6.) Line 136: is it really unexpected? I suggest to phrase it a bit more careful and maybe move this sentence a bit later in the discussion, not as an opener.
7.) I know the word count is limited, however, if infection takes place or not depends not only on ACE2; TMPRRSS2 and other factors also contribute, is it worth to add this in the discussion? it sounds like that ACE2 is the one and only factor.
8.) I am not a native English speaker, however, please replace the word "till" with "until" or use a "-"
9.) can you please include a bar to the immunocytochemistry?
10.) picture B has a very mild counterstain which makes it difficult to appreciate the cellular/nuclear morphology, do you have one with a stronger signal?
11.) is there any information about another route of entry than pool water? e.g. food etc.? has this been elaborated?
Looking forward to reading the revised version of your manuscript.
Wishing you a great new year and all the best for 2023.
Round 2
Reviewer 2 Report
The Authors have addressed my major concerns.